

# Selection of suitable wheat genotypes under thermal stress and complex genotype-environment interaction using stability analyses and selection indices

Abdelhalim Ghazy[1], Walid Ben Romdhane[1], Majed Alotaibi[1], Abdullah Al-Doss[1], Omar Dahrog[1], Nasser Al-Suhaibani[1], Abdullah Ibrahim[1], Adel M. Al-Saif[1], Khalid A. Al-Gaadi[2], Ahmed M. Zeyada[2], Khalid F. Almutairi[1] and Ibrahim Al-Ashkar[1]

[1] Department of Plant Production, King Saud University, Riyadh, Saudi Arabia
[2] Department of Agricultural Engineering, King Saud University, Riyadh, Saudi Arabia

Corresponding author
Ibrahim Al-Ashkar,
ialashkar@ksu.edu.sa

## ABSTRACT

Thermal stress is a consequence of climate change that threatens food security, causes plant tissue damage, and harms crop production, particularly during the pollination and fertilization period and in grain-filling stages negatively impacting the number of grains, grain size, and quality. Genotype-environment interaction (GEN: ENV) complicates the selection of optimal wheat genotypes due to the complex genetic basis of yield under varying conditions. Diversified approaches were put forth in response to the pressing demand for simultaneous enhancements in high-yield performance combined with stability. This study investigates the selection of ideal wheat genotypes under thermal stress and complex GEN: ENV using stability analyses and selection indices to assess genotype performance and stability. Twenty wheat genotypes were evaluated across optimal conditions (OC) and thermal stress conditions (TSC) over three growing seasons with six ENVs. Results demonstrated significant GEN: ENV, revealing genetic variations in thermal tolerance. The additive main effects and multiplicative interaction (AMMI2) biplot indicated a combined variance of 99.00%, and eleven genotypes showed stable grain yield (GY) with six ENVs, three (G05, G09, and G17) were more stable. The G04, G05, G06, G09, and G18 genotypes were chosen for GY as perfect (stable and high-performance) genotypes by weighted average of absolute scores biplot (WAASB) and were also identified as the best genotypes group by WAASB-GY, with the exception of G18. Ten selection indices showed significant positive associations under $GY_{oc}$ and $GY_{tsc}$, so they can be leveraged to detect the genotype's high yield of $GY_{tsc}$ indirectly. The heritability, accuracy, and $r_{gen: env}$ values for most indices were high, indicating a major role of the genotypic effect in their inheritance, with the exception of the stress-non-stress production index (SNPI) index. Out of the five that were examined by WAASB, G04, G05, G06, and G09 were the top-ranking genotypes by the multi-trait genotype ideotype distance index, either before or after removing variables. This suggests that they could be examined for validation stability measures. The findings of this study offer valuable insights for ENVs variety selection, facilitating the identification of improved cultivars and supporting the development of thermal stress-resilient breeding programs.

## INTRODUCTION

The rate of dietary consumption continues to rise steadily due to an annual birth rate increase of 1.1% on average—far exceeding many experts' predictions (*Farhad et al., 2023*)—alongside a concurrent agricultural revolution. Despite significant advancements in the area of agriculture to support global nutritional needs, these advances are increasingly vulnerable to instability and continuing threats due to negative climate change (*Motawei, Kamara & Rehan, 2025*). Heat stress is a consequence of climate change that harms crop production, particularly during the pollination and fertilization period (transfer of the pollen across the pollen tube to the ovary which negatively affects the number of grains) and grain-filling stages (negatively affecting grain size and quality). It poses a threat to food security and is predicted to have an increasingly severe and negative impact on the amount of wheat produced over time as global temperatures rise (*Al-Ashkar et al., 2020*; *Farhad et al., 2023*). Due to the strong inverse relation between high seasonal temperatures and crop yields, a significant portion of global agricultural yield loss is attributed to crop tissue damage caused by thermal stress (*Akter & Islam, 2017*; *Farhad et al., 2023*; *Gammans, Mérel & Ortiz-Bobea, 2017*; *Suzuki et al., 2012*). To accomplish this goal, more research is needed on increasing crop heat tolerance to meet global food needs. International food policy must give priority to ensuring global food security by stimulating and encouraging scholars to initiate research collaboration to produce unique heat stress-tolerant wheat genotypes (*Al-Ashkar et al., 2020*; *Al-Ashkar et al., 2023b*; *Arif et al., 2025*).

Due to global climate change, the National Oceanic and Atmospheric Administration (*National Oceanic and Atmospheric Administration (NOAA), 2023*) notes a trend toward warmer winters across expanding regions, alongside a consistently heightened greenhouse effect worldwide. This is undesirable as warmer conditions adversely affect winter crops, impacting many yield-contributing traits such as grain numbers, grain size, and grain weight (*Al-Ashkar et al., 2020*; *Fu et al., 2023*; *Poudel & Poudel, 2020*). Based on the data for field yield and weather of several regional scales to know the effects of high temperature on wheat productivity, a 1 °C increase in mean air temperature in the growing season was estimated to reduce wheat yield by 3–21% (*Barkley et al., 2014*; *Fu et al., 2023*; *Lobell et al., 2005*; *Tiwari et al., 2013*; *You et al., 2009*). Globally, wheat yield loss is estimated at 6.0 ± 2.9% for each 1 °C increase (*Arif et al., 2025*; *Asseng et al., 2014*; *Fu et al., 2023*; *Zhao et al., 2017*). A plant's ability to overcome thermal stress depends on appropriate environmental conditions, agronomic practices, and genetic factors that enhance evaporative cooling potential (*Braun, Atlin & Payne, 2010*). A sustainable approach to reducing heat stress damage involves developing tolerant varieties by examining various genotypes to identify those with tolerance and then transferring these traits into commercially cultivated varieties to obtain high-yielding model varieties, combining productivity and thermal tolerance (*Al-Ashkar et al., 2023b*; *Fu et al., 2023*).
Heat is a polygenic trait that makes it highly vulnerable to the environment, so it is important to evaluate the genotypes under heat stress (genotype performance varies from superior to inferior or vice-versa across different seasons) to determine which ones are ideal for choosing as commercial varieties (genetic stable for sites/site) to become gratifying for the farmers and/or will be introduced in prospective breeding programs for continual improvement (*Al-Ashkar et al., 2020*; *Hamidou, Halilou & Vadez, 2012*; *Qaseem et al., 2018*; *Singamsetti et al., 2021*), since a quantitative trait like grain yield has very little heritability (*Saba et al., 2001*). From this point of view, several heat tolerance indices have been proposed building on the mathematical relation of genotype yielded capacity under non-stress and heat stress to measure the level of tolerance and select the heat tolerant genotypes (*Bennani et al., 2017*; *Lamba et al., 2023*). A reliable heat tolerance index must be able to distinguish genotypes and determine the best ones under non-stress and heat stress (*Bennani et al., 2017*; *Saba et al., 2001*). However, the effectiveness of selection indices in distinguishing tolerant genotypes depends on the intensity of environmental stress, which varies across years and regions, thereby affecting the efficacy of selection indices in identifying tolerant genotypes, so, the genotypes that exhibit exceptional performance over various stress intensities ought to be chosen (*Bennani et al., 2017*; *Farshadfar et al., 2012*; *Lamba et al., 2023*). *Bennani et al. (2017)* indicated that while multiple studies have highlighted the efficiency of selection indices for tolerance, these studies did not fully address it due to the indices' dependence on simple statistics.

The application of multivariate statistical methods has accuracy in the successful selection of genotypes in breeding programs by combining all studied variables at once. This integrated method based on highly computationally capable models of multidimensional data may provide a better understanding of breeding programs, which may help identify favorable genotypes (*Abdolshahi et al., 2015*; *Al-Ashkar et al., 2019*; *Al-Ashkar et al., 2022*; *Chakraborty et al., 2020*; *Salami et al., 2025*). Therefore, multivariate statistical methods such as analyses of principal component (PCA) are used to select the most crucial variables and minimize the number of them, cluster to collect performance convergent genotypes with each other, discriminant to strengthen the credibility of clustering, additive main-effects and multiplicative interaction (AMMI) to predict for genotype × environment, multi-trait genotype-ideotype distance index (MGIDI) to detect ideotype as it focuses on selecting the genotype depending on multiple traits (with its ability to assess the strengths and weakness of the selected genotypes), and the weighted average of absolute scores (WAASB) index to recognize the high-yielding and stable genotypes, could serve as models for screening tests and for identifying the sources of variation (*Al-Ashkar et al., 2022*; *Farhad et al., 2022*; *Olivoto et al., 2019a*; *Olivoto & Nardino, 2021*; *Salami et al., 2025*).

Since the AMMI analysis was one of the best models used for the selection of preference genotypes offers a lot of advantages in interpreting genotype-environment interaction (GEN: ENV), a main limitation was noted when analyzing the structure of the linear mixed-effect model (LMM), therefore, a novel model, referred as weighted average of absolute scores (WAASB), was proposed by *Olivoto et al. (2019a)*. WAASB resulted from the singular value decomposition of BLUP (best linear unbiased prediction) matrix for GEN: ENV effects generated by an LMM for the description of greater ideal genotypes based

**Table 1  Environment code used and experiments description for production environments.**

| Environment code | Experiments | Planting dates | Season |
|---|---|---|---|
| ENV1 | Optimal conditions (timely sown) | 15 November | 2018/19 |
| ENV2 | Thermal stress conditions (late sown) | 20 December | 2018/19 |
| ENV3 | Optimal conditions (timely sown) | 17 November | 2019/20 |
| ENV4 | Thermal stress conditions (late sown) | 25 December | 2019/20 |
| ENV5 | Optimal conditions (timely sown) | 17 November | 2020/21 |
| ENV6 | Thermal stress conditions (late sown) | 25 December | 2020/21 |

on a combination of stability and yield performance (*Olivoto et al., 2019a*). The WAASB model combines the characteristic features of the AMMI and BLUP models (as distinct approaches achieving the same goal of discriminating the GEN: ENV pattern from random error, despite being statistically different) in a unique one index, allowing the selection of high-yielding and stable genotypes (*Ahakpaz et al., 2021*; *Al-Ashkar et al., 2023a*; *Zuffo et al., 2020*). In this perspective, the present study aimed to (i) identify the optimal genotypes that combine stability and high productivity to confront thermal stress (ii) validate the proficiency of 18 selection indices used in screening tolerant genotypes *via* a variety of statistical approaches (iii) assess the associations among the different indices.

# MATERIALS AND METHODS

## Experiment description

Experimental material: Twenty wheat genotypes were chosen (DHL12 (G01), DHL02 (G02), DHL25 (G03), DHL07 (G04), DHL26 (G05), Gemmeiza-9 (G06), DHL11 (G07), KSU106 (G08), Gemmeiza-12 (G09), DHL01 (G10), DHL14 (G11), DHL29 (G12), DHL15 (G13), DHL06 (G14), Misr1 (G15), DHL05 (G16), DHL23 (G17), Sakha-93 (G18), Pavone-76 (G19) and DHL08 (G20)), the pedigree for these genotypes is listed in Table S1. Environment description: The experiment was conducted for three seasons from 2018/19 to 2020/21 at the King Saud University Agricultural Research Station (24°42'N, 44°46'E, 400 m asl), with a total of six experiments/environments (ENVs), the environments (optimal conditions (OC) and thermal stress conditions (TSC)) were separated (Table 1). Each environment for twenty genotypes was three-repeated in a randomized complete block design. Plot area, texture soil type, seedling rate, fertilizer rates and the timing of their application, and meteorological conditions (Table S2) as detailed in earlier studies (*Al-Ashkar et al., 2022*; *Al-Ashkar et al., 2023c*).

## Measurements

To measure differences between the 20 genotypes used under (OC and TSC), the grain yield (GY, ton ha$^{-1}$) trait was valuated after harvest from yield three rows two m long. The GY data had been used to assess heat tolerance indices according to the subsequent mathematical formulas presented by *Bennani et al. (2017)* and *Lamba et al. (2023)*. $TOL_{stress\ tolerance} = GY_{oc} - GY_{tsc}$, $STI_{stress\ tolerance\ index} = (GY_{oc} \times GY_{tsc})/\overline{x}_{tsc}^2$, $STI_{m\ modified\ stress\ tolerance\ index} = [(\Sigma GY_{tsc}^2/\Sigma \overline{GY}_{tsc}^2) \times STI]$, $SSI_{stress\ susceptibility\ index} = [(1 - (GY_{tsc}/GY_{tsc}))/(1 - (\overline{x}_{tsc}/\overline{x}_{oc}))]$, $SSPI_{stress\ susceptibility\ percentage\ index} = [(GY_{oc} -$

$GY_{tsc}/2\overline{x}_{oc}) \times 100]$, $YI_{\text{yield index}} = [GY_{tsc}/\overline{x}_{oc}]$, $YSI_{\text{yield stability index}} = [GY_{tsc}/GY_{oc}]$, $RDI_{\text{relative drought index}} = [(GY_{tsc}/GY_{oc})/(\overline{x}_{tsc}/\overline{x}_{oc})]$, $MP_{\text{mean productivity}} = [(GY_{oc}+GY_{tsc})/2]$, $GMP_{\text{geometric mean productivity}} = [\sqrt{(GY_{oc} \times GY_{tsc})}]$, $HM_{\text{harmonic mean}} = 2[(GY_{oc} \times GY_{tsc})/(GY_{oc} + GY_{tsc})]$, $MRP_{\text{mean relative performance}} = [(GY_{tsc}/\overline{x}_{tsc}) + (GY_{tsc}/\overline{x}_{oc})]$, $PYR_{\text{percent yield reduction}} = [((GY_{oc} - GY_{tsc})/GY_{oc}) \times 100]$, $REI_{\text{relative efficiency index}} = (GY_{tsc}/\overline{x}_{tsc}) \times (GY_{oc}/\overline{x}_{oc})$, $ATI_{\text{abiotic tolerance index}} = (GY_{oc} - GY_{tsc})/(\overline{x}_{oc}/\overline{x}_{tsc}) \times \sqrt{((GY_{oc} \times GY_{tsc})}$, $SNPI_{\text{stress/non-stress production index}} = \left[\sqrt[3]{(GY_{oc} + GY_{tsc})/(GY_{oc} - GY_{tsc})}\right] \times \left[\sqrt[3]{(GY_{oc} \times GY_{tsc} \times GY_{tsc})}\right]$, $SWPI_{\text{stress-weighted performance index}} = \sqrt{GY_{oc}}/GY_{tsc}$ and $RSC_{\text{relative stress change}} = ((GY_{oc} - GY_{tsc})/GY_{oc}) \times 100$, where $GY_{oc}$ and $GY_{tsc}$ are the GY of genotypes, while $\overline{x}_{oc}$ and $\overline{x}_{tsc}$ are the overall mean GY under optimal conditions (oc) and thermal stress conditions (tsc), respectively.

## Statistical analyses

The variance components were appreciated by restricted maximum likelihood (REML) as described by *Dempster, Laird & Rubin (1977)*. To evaluate the significance of the random effects, a likelihood ratio test (LRT) was performed involving comparing two models (one that included all random terms and another that excluded one of these terms), utilizing a chi-square ($\chi^2$) test for the comparison. Eight parameters were calculated as described by *Sampaio Filho et al. (2023)*:

- Heritability$_{\text{expected mean square}}$ $(h^2_{ems}) = (\sigma^2_{gen})/(\sigma^2_{gen} + \frac{\sigma^2_{gen:env}}{b} + \sigma^2_{res})$
- Heritability$_{\text{plot mean}}$ $(h^2_{pm}) = (\sigma^2_{gen})/(\sigma^2_{gen} + \frac{\sigma^2_{gen:env}}{b \times env} + \frac{\sigma^2_{res}}{b \times env})$
- Accuracy $= \sqrt{h^2_{pm}}$
- Coefficient of determination$_{\text{GEN:ENV effects}}$ $(R^2) = (\sigma^2_{gen:env})/(\sigma^2_{gen} + \sigma^2_{gen:env} + \sigma^2_{res})$
- Coefficient of variation$_{\text{genotypic}}$ $(CV_{gen}) = \sqrt{\sigma^2_{gen}/\overline{x}} \times 100$
- Coefficient of variation$_{\text{residual}}$ $(CV_{res}) = \sqrt{\sigma^2_{res}/\overline{x}} \times 100$
- $CV$ ratio $= Cv_{gen}/Cv_{res}$
- Correlation$_{\text{genotype-environment}}$ $(r_{gen:env}) = (\sigma^2_{gen})/(\sigma^2_{gen} + \sigma^2_{gen:env})$

where $\sigma^2_{gen}$, $\sigma^2_{gen:env}$ and $\sigma^2_{res}$ signify the variances of genotypic, genotype × environment, and residual (error), respectively; b and env signify the blocks number and environments respectively $\overline{x}$ is the overall mean.

Data of GY trait from six ENVs underwent a variety of analyses for estimating genetic stability—AMMI analysis (AMMI-ANOVA and AMMI biplots; AMMI's model was employed to assess multiplicative effects and identify stable genotypes), Joint regression model, stability indexes, and WAAS biplot). Data of selection indices generated by GY under optimal and thermal stress conditions underwent a variety of analyses for estimating relationships between the various indices, including genetic (rg) and phenotypic (rp) correlations, genetic parameters, and MGIDI index. All statistics analyses and biplots were created by RStudio packages (R version 4.3.3; *R Core Team, 2023*). The metan R package was used as per *Olivoto, Lúcio & Jarman (2020)*. Selection indices were computed to the mathematical formulas by Microsoft Excel 2019.
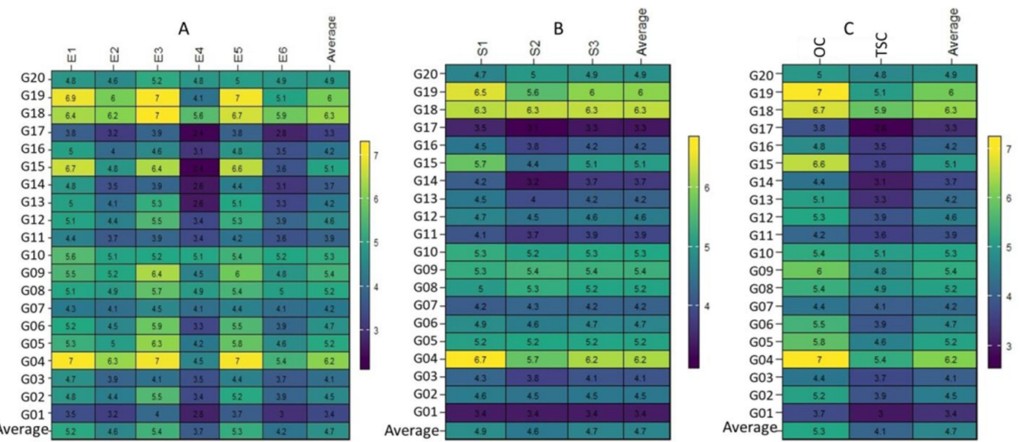

**Figure 1** Plotting the mean performance for 20 wheat genotypes.

## RESULTS

### The variance in wheat grain yield

This conclusion was strengthened by the range in the performance of the genotypes assessed, which varied for optimum conditions from 3.5 (t ha$^{-1}$) (G01 in E1) to 7.0 (t ha$^{-1}$) (G04 in E1), (G04, G18 and G19 in E3) and (G04, and G19 in E5). The G01 genotype showed the minimum performance at one place (E1). In contrast, the G04 genotype showed the maximum performance at the three ENVs (E1, E3, and E5). In every optimum condition, genotype G04 performed best (Fig. 1A). Under thermal stress values, they varied from 2.4 (t ha$^{-1}$) (G15 and G17 in E4) to 6.3 (t ha$^{-1}$) (G04 in E2). The G15 and G17 genotypes showed the lowest performance at one place (E4), whereas the G04 genotype showed the highest performance at one place (E2) (Fig. 1A). The performance of the genotypes in the three seasons varied from 3.1 (t ha$^{-1}$) (G17 in S2) to 6.7 (t ha$^{-1}$) (G04 in S1), and genotypes G04 or G18 performed best in the three seasons and the average (Fig. 1B). In the case of treatments, the values ranged from 3.7 (t ha$^{-1}$) in (G04) to 7.0 (t ha$^{-1}$) in (G04 and G17) under OC, and ranged from 2.8 (t ha$^{-1}$) in (G17) to 5.9 (t ha$^{-1}$) in (G18) under TSC. In the two cases, genotypes G04 and G18 grossed the most (Fig. 1C).

### Joint ANOVA and AMMI model analyses for grain yield

The joint analysis of variance (ANOVA) and AMMI model for the six environments is shown in Table 2. The joint ANOVA determined that the GEN, ENV, and GEN: ENV were highly significant (Table 2), given that GEN: ENV significantly impacts GY. IPCA [1] and IPCA [2] were determined to be significant, and there was an inequality between ENVs for genotype classifications. The best-predicted AMMI model was with two IPCs, the first two components were significant and accounted for 84.90% and 14.10% of the GEN: ENV, respectively, for six ENVs at the 0.001 probability level.

**Table 2  AMMI analysis of variance for grain yield trait among 20 genotypes in six environments.**

| Source | df | SS | MS | F-Value | Total variation explained (%) | | GEN × ENV variation explained (%) | |
|---|---|---|---|---|---|---|---|---|
| | | | | | Proportion | Accumulated | Proportion | Accumulated |
| ENV | 5 | 138.00 | 27.50 | 335.000[***] | 25.81 | 25.81 | | |
| REP(ENV) | 12 | 0.99 | 0.08 | 1.690[ns] | 0.18 | 25.99 | | |
| GEN | 19 | 264.00 | 13.90 | 286.000[***] | 49.52 | 75.51 | | |
| GEN:ENV | 95 | 59.90 | 0.63 | 13.000[***] | 11.24 | 86.75 | | |
| IPCA[1] | 23 | 50.90 | 2.21 | 45.600[***] | 9.55 | 96.3 | 84.90 | 84.90 |
| IPCA[2] | 21 | 8.46 | 0.40 | 8.300[***] | 1.59 | 97.89 | 14.10 | 99.00 |
| IPCA[3] | 19 | 0.53 | 0.03 | 0.580[ns] | 0.10 | 97.99 | 0.90 | 99.90 |
| IPCA[4] | 17 | 0.00 | 0.00 | 0.050[ns] | 0.00 | 97.99 | 0.10 | 100.00 |
| IPCA[5] | 15 | 0.00 | 0.00 | 0.000[ns] | 0.00 | 97.99 | 0.00 | 100.00 |
| Residuals | 228 | 11.10 | 0.05 | | 2.01 | 100.00 | | |
| Total | 454 | 533.00 | 1.17 | | | | | |

**Notes.**
df, Degrees of freedom; SS, Sum of squares; MS, mean squares.
[***]Significant at 0.001.
[ns]not significant.

## Joint regression model of stability analysis

The joint regression model (*Eberhart & Russell, 1966*) detected highly significant differences by a pooled ANOVA for all model effects (Table 3). The mean GY ranged between 3.30 (G17) to 6.31 (G18), with an average of 4.72 t ha$^{-1}$. The stability analysis parameter (bi) noted no genotype had bi = 1 and $S^2$di = 0. The genotypes G05 and G09 had $b_i$ values close to 1 indicating that they are more stable under every six ENVs (Table 3). Genotypes G04 ($\mu$= 6.20, bi = 1.51***, $S^2$di = 0.042***), G06 ($\mu$= 4.74, bi = 1.420***, $S^2$di = 0.011), G12 ($\mu$= 4.59, bi = 1.240***, $S^2$di = −0.008), G13 ($\mu$= 4.24, bi = 1.610***, $S^2$di = −0.014), G15 ($\mu$= 5.07, bi = 2.650***, $S^2$di = 0.053***) and G19 ($\mu$= 6.04, bi = 1.750***, $S^2$di = 0.020) were observed to be stable in optimal (ENV1, ENV3 and ENV5) conditions (Table 3), whereas for genotypes G08 ($\mu$= 5.20, bi = 0.393**, $S^2$di = 0.031*), G10 ($\mu$= 5.25, bi = 0.161**, $S^2$di = 0.017), and G18 ($\mu$= 6.31, bi = 0.735**, $S^2$di = 0.019), high means with bi values less than 1 indicate that these genotypes show more resilience to unfavorable environments as thermal stress (ENV2, ENV4 and ENV6) were observed. The root mean square error (RMSE) is used to evaluate the prediction quality, which ranged between 0.021 (G17) and 0.327 (G14), while $R^2$ values ranged between 0.265 (G20) and 0.999 (G13 and G17).

## Stability indexes of evaluated genotypes

The Annicchiarico method measures genotypic stability, which received the top rank for genotypes G19, G04, G18, and G15 of analysis favorable environment, genotypes G18, G04, G10, and G19 of analysis unfavorable environment, and genotypes G18, G04, G19, and G09 of general analysis (Table 4). Shukla's rank-sum method integrates mean performance and stability into a unified selection criterion, which revealed that the top four ranks were for genotypes (G17, G12, G02, and G16), which matched in ranking with *Wricke*'s (*1962*) ecovalence. The AMMI-based stability parameter (ASTAB) computes by significant

**Table 3  Pooled analysis of variance of 20 wheat genotypes across six environmental for GY (*Eberhart & Russell, 1966* model).**

| S.O.V | | Df | MS | *F* value | Pr (>F) |
|---|---|---|---|---|---|
| GEN | | 19 | 13.900 | 104.511 | 0.000 |
| ENV + (GEN × ENV) | | 100 | 1.980 | 14.887 | 0.000 |
| ENV (linear) | | 1 | 138.000 | 1037.594 | 0.000 |
| GEN x ENV (linear) | | 19 | 2.590 | 19.474 | 0.000 |
| Pooled deviation | | 80 | 0.133 | 2.742 | 0.000 |
| Pooled error | | 228 | 0.049 | | |
| Stability parameters | | | | | |
| GEN | GY | $b_i$ | $s^2$ di | RMSE | $R^2$ |
| G01 | 3.36 | 0.653[***] | 0.002 | 0.109 | 0.932 |
| G02 | 4.54 | 1.170[*] | 0.016 | 0.147 | 0.961 |
| G03 | 4.06 | 0.577[***] | 0.052[***] | 0.212 | 0.739 |
| G04 | 6.20 | 1.510[***] | 0.042[***] | 0.198 | 0.957 |
| G05 | 5.20 | 1.050[ns] | 0.053[***] | 0.214 | 0.902 |
| G06 | 4.74 | 1.420[***] | 0.011 | 0.135 | 0.977 |
| G07 | 4.24 | 0.256[***] | −0.004 | 0.091 | 0.753 |
| G08 | 5.20 | 0.393[***] | 0.031[*] | 0.177 | 0.652 |
| G09 | 5.39 | 1.010[ns] | 0.061[***] | 0.227 | 0.884 |
| G10 | 5.25 | 0.161[***] | 0.017 | 0.148 | 0.312 |
| G11 | 3.87 | 0.490[***] | 0.030[*] | 0.175 | 0.750 |
| G12 | 4.59 | 1.240[***] | −0.008 | 0.072 | 0.991 |
| G13 | 4.24 | 1.610[***] | −0.014 | 0.038 | 0.999 |
| G14 | 3.72 | 1.110[ns] | 0.144[***] | 0.327 | 0.815 |
| G15 | 5.07 | 2.650[***] | 0.053[***] | 0.214 | 0.983 |
| G16 | 4.17 | 1.110[ns] | 0.032[*] | 0.180 | 0.936 |
| G17 | 3.30 | 0.937[ns] | −0.016 | 0.021 | 0.999 |
| G18 | 6.31 | 0.735[***] | 0.019 | 0.153 | 0.898 |
| G19 | 6.04 | 1.750[***] | 0.020 | 0.156 | 0.980 |
| G20 | 4.89 | 0.157[***] | 0.023[*] | 0.162 | 0.265 |

**Notes.**
[*] Significant at 0.05.
[***] Significant at 0.001.
[ns] not significant.

interaction principal components (IPCs) in the AMMI model, which revealed that the top four ranks were genotypes G17, G12, G01, and G02. AMMI Stability Index (ASI), AMMI-stability value (ASV), modified AMMI Stability Index (MASI), modified AMMI Stability value (MASV), and weighted average of absolute scores (WAAS) were matched in ranking genotypes, which received the top four genotypes G17, G05, G09, and G16. Annicchiarico's D parameter values (DA) and stability measure based on fitted AMMI model (FA) were matched in ranking genotypes, which received the top four genotypes G17, G12, G02, and G16. Zhang's D parameter (DZ) and sums of the averages of the squared eigenvector values (EV) were matched in ranking genotypes, which received the top four genotypes G17, G12, G01, and G18. Sums of the absolute value of the IPC Scores

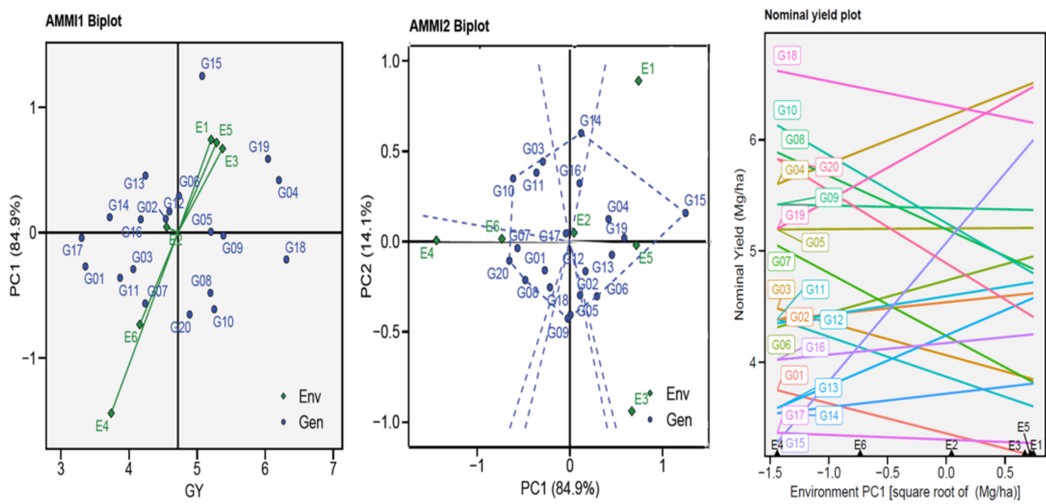

**Figure 2** AMMI1, AMMI2 and nominal biplot for the GY trait of 20 wheat genotypes evaluated in six environments.

(SIPC), which received the top four genotypes G17, G12, G02, and G05. WAASY (the index that considers the weights for stability and productivity in the genotype ranking) received the top four genotypes G18, G09, G04, and G05 (Table 4).

## Analyses biplots
### AMMI biplot

The AMMI biplots (GGE biplots) were used to visually the yield potential (GY) of the twenty genotypes in six environments by describing the relationship between the genotypes and six environments *vs.* IPC1 (Fig. 2). The AMMI1 biplot indicated that the one ENV4 (E4) was beyond their sources and had a longer vector, pointing to a higher interaction, while the five other ENVs had a shorter vector and were closer to their source point to a lower interaction. The non-stressful environments (E1, E3, and E5) had an angle between the vectors mostly less than 90°, pointing to positive correlations, and the heat-stress environments (E2, E4, and E6) gave themselves similar results. This suggests that when applied under comparable conditions, the genotype–environment interactions (GEI) effects are often given within the same range. The AMMI2 biplot indicated that IPCA1 and IPCA2 described a combined variance of 99.00%. Figure 2 shows the GEI volume with the ENV type by the vertical projection from the GEN to the ENV vector. Accordingly, the genotypes might be viewed (G03, G05, G06, G08, G09, G10, G14, G15, and G20) as unstable in environments used. Using the AMMI biplot (generic genotypic adaptation) map (nominal plot) for genotypes. The adaptation map showed that G05, G09, and G17 were more suited, and exhibited identical performance in all environments. Although they vary from environment to environment, G15 performed best in E1, unlike G10 and G20 performance of the least in E1. The G02, G11, and G16 exhibited reduced GEI.

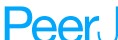

**Table 4  Stability indexes of 20 wheat genotypes across six environmental for GY.**

| GEN | Annichiarico environment index | | | | | | Shukla | | ecovalence | | ASTAB | | ASI | | ASV | |
| | Favorable | | Unfavorable | | General | | | | | | | | | | | |
| | value | rank | value | rank | value | rank | value | rank | value | rank | value | rank | value | rank | value | rank |
|---|---|---|---|---|---|---|---|---|---|---|---|---|---|---|---|---|
| G01 | 70.50 | 20 | 72.10 | 17 | 71.30 | 19 | 0.065 | 8 | 1.040 | 8 | 0.098 | 3 | 0.230 | 9 | 1.630 | 9 |
| G02 | 98.20 | 12 | 93.40 | 10 | 95.80 | 11 | 0.033 | 3 | 0.596 | 3 | 0.101 | 4 | 0.102 | 5 | 0.725 | 5 |
| G03 | 83.70 | 16 | 89.50 | 13 | 86.60 | 15 | 0.139 | 11 | 2.040 | 11 | 0.281 | 14 | 0.255 | 11 | 1.810 | 11 |
| G04 | 132.00 | 2 | 130.00 | 2 | 131.00 | 2 | 0.173 | 13 | 2.490 | 13 | 0.190 | 10 | 0.355 | 13 | 2.520 | 13 |
| G05 | 110.00 | 6 | 111.00 | 7 | 110.00 | 5 | 0.051 | 5 | 0.845 | 5 | 0.165 | 7 | 0.058 | 2 | 0.408 | 2 |
| G06 | 104.00 | 7 | 94.70 | 12 | 99.60 | 9 | 0.101 | 9 | 1.520 | 9 | 0.178 | 8 | 0.252 | 10 | 1.790 | 10 |
| G07 | 83.20 | 15 | 98.80 | 9 | 91.00 | 14 | 0.281 | 16 | 3.950 | 16 | 0.320 | 15 | 0.479 | 16 | 3.400 | 16 |
| G08 | 103.00 | 8 | 120.00 | 6 | 111.00 | 6 | 0.218 | 15 | 3.100 | 15 | 0.277 | 13 | 0.409 | 15 | 2.900 | 15 |
| G09 | 113.00 | 5 | 116.00 | 5 | 114.00 | 4 | 0.057 | 7 | 0.926 | 7 | 0.182 | 9 | 0.063 | 3 | 0.448 | 3 |
| G10 | 102.00 | 10 | 124.00 | 3 | 113.00 | 7 | 0.376 | 18 | 5.230 | 18 | 0.496 | 19 | 0.521 | 18 | 3.700 | 18 |
| G11 | 79.00 | 18 | 86.20 | 15 | 82.60 | 17 | 0.162 | 12 | 2.340 | 12 | 0.275 | 12 | 0.311 | 12 | 2.200 | 12 |
| G12 | 100.00 | 9 | 93.60 | 11 | 96.80 | 10 | 0.025 | 2 | 0.491 | 2 | 0.056 | 2 | 0.145 | 7 | 1.030 | 7 |
| G13 | 97.20 | 11 | 79.80 | 16 | 88.50 | 16 | 0.179 | 14 | 2.580 | 14 | 0.211 | 11 | 0.385 | 14 | 2.730 | 14 |
| G14 | 82.80 | 17 | 73.50 | 18 | 78.10 | 18 | 0.137 | 10 | 2.010 | 10 | 0.375 | 17 | 0.134 | 6 | 0.950 | 6 |
| G15 | 124.00 | 4 | 85.10 | 19 | 105.00 | 12 | 1.430 | 20 | 19.500 | 20 | 1.580 | 20 | 1.060 | 20 | 7.510 | 20 |
| G16 | 91.20 | 14 | 84.80 | 14 | 88.00 | 13 | 0.038 | 4 | 0.670 | 4 | 0.115 | 6 | 0.100 | 4 | 0.711 | 4 |
| G17 | 72.60 | 19 | 66.60 | 20 | 69.60 | 20 | −0.009 | 1 | 0.035 | 1 | 0.004 | 1 | 0.037 | 1 | 0.259 | 1 |
| G18 | 127.00 | 3 | 143.00 | 1 | 135.00 | 1 | 0.055 | 6 | 0.905 | 6 | 0.110 | 5 | 0.186 | 8 | 1.320 | 8 |
| G19 | 132.00 | 1 | 122.00 | 4 | 127.00 | 3 | 0.310 | 17 | 4.350 | 17 | 0.345 | 16 | 0.498 | 17 | 3.530 | 17 |
| G20 | 94.70 | 13 | 116.00 | 8 | 105.00 | 8 | 0.385 | 19 | 5.360 | 19 | 0.437 | 18 | 0.554 | 19 | 3.930 | 19 |

| GEN | DA | | DZ | | EV | | FA | | MASI | | MASV | | SIPC | | WAAS | | WAASY | |
| | value | rank | value | rank | value | rank | value | rank | value | rank | value | rank | value | rank | value | rank | value | rank |
|---|---|---|---|---|---|---|---|---|---|---|---|---|---|---|---|---|---|---|
| G01 | 0.585 | 8 | 0.181 | 3 | 0.016 | 3 | 0.343 | 8 | 0.230 | 9 | 1.630 | 9 | 0.429 | 6 | 0.254 | 9 | 40.90 | 19 |
| G02 | 0.445 | 3 | 0.236 | 7 | 0.028 | 7 | 0.198 | 3 | 0.102 | 5 | 0.725 | 5 | 0.407 | 3 | 0.137 | 5 | 66.10 | 6 |
| G03 | 0.824 | 11 | 0.371 | 17 | 0.069 | 17 | 0.679 | 11 | 0.255 | 11 | 1.810 | 11 | 0.734 | 16 | 0.313 | 11 | 49.70 | 15 |
| G04 | 0.863 | 12 | 0.227 | 5 | 0.026 | 5 | 0.745 | 12 | 0.355 | 13 | 2.520 | 13 | 0.542 | 10 | 0.376 | 13 | 82.30 | 3 |
| G05 | 0.527 | 5 | 0.314 | 13 | 0.049 | 13 | 0.278 | 5 | 0.058 | 2 | 0.408 | 2 | 0.413 | 4 | 0.063 | 2 | 80.60 | 4 |
| G06 | 0.712 | 9 | 0.275 | 9 | 0.038 | 9 | 0.507 | 9 | 0.252 | 10 | 1.790 | 10 | 0.596 | 11 | 0.294 | 10 | 61.80 | 9 |
| G07 | 1.150 | 16 | 0.280 | 10 | 0.039 | 10 | 1.320 | 16 | 0.479 | 16 | 3.400 | 16 | 0.601 | 12 | 0.489 | 16 | 44.20 | 17 |
| G08 | 1.010 | 15 | 0.289 | 11 | 0.042 | 11 | 1.030 | 15 | 0.409 | 15 | 2.900 | 15 | 0.694 | 14 | 0.443 | 15 | 62.40 | 8 |
| G09 | 0.555 | 7 | 0.329 | 14 | 0.054 | 14 | 0.308 | 7 | 0.063 | 3 | 0.448 | 3 | 0.449 | 7 | 0.080 | 3 | 82.80 | 2 |
| G10 | 1.320 | 18 | 0.404 | 18 | 0.082 | 18 | 1.740 | 18 | 0.521 | 18 | 3.700 | 18 | 0.960 | 19 | 0.574 | 18 | 57.10 | 11 |
| G11 | 0.883 | 13 | 0.344 | 16 | 0.059 | 16 | 0.779 | 13 | 0.311 | 12 | 2.200 | 12 | 0.742 | 17 | 0.363 | 12 | 44.10 | 18 |
| G12 | 0.403 | 2 | 0.152 | 2 | 0.012 | 2 | 0.163 | 2 | 0.145 | 7 | 1.030 | 7 | 0.333 | 2 | 0.168 | 6 | 65.50 | 7 |
| G13 | 0.926 | 14 | 0.231 | 6 | 0.027 | 6 | 0.857 | 14 | 0.385 | 14 | 2.730 | 14 | 0.528 | 9 | 0.400 | 14 | 48.60 | 16 |
| G14 | 0.816 | 10 | 0.467 | 19 | 0.109 | 19 | 0.666 | 10 | 0.134 | 6 | 0.950 | 6 | 0.722 | 15 | 0.190 | 7 | 49.80 | 14 |
| G15 | 2.540 | 20 | 0.627 | 20 | 0.196 | 20 | 6.460 | 20 | 1.060 | 20 | 7.510 | 20 | 1.410 | 20 | 1.090 | 20 | 29.40 | 20 |
| G16 | 0.470 | 4 | 0.254 | 8 | 0.032 | 8 | 0.221 | 4 | 0.100 | 4 | 0.711 | 4 | 0.428 | 5 | 0.136 | 4 | 60.00 | 10 |
| G17 | 0.104 | 1 | 0.041 | 1 | 0.001 | 1 | 0.011 | 1 | 0.037 | 1 | 0.259 | 1 | 0.087 | 1 | 0.043 | 1 | 50.00 | 13 |
| G18 | 0.546 | 6 | 0.223 | 4 | 0.025 | 4 | 0.298 | 6 | 0.186 | 8 | 1.320 | 8 | 0.468 | 8 | 0.220 | 8 | 91.60 | 1 |
| G19 | 1.190 | 17 | 0.289 | 12 | 0.042 | 12 | 1.420 | 17 | 0.498 | 17 | 3.530 | 17 | 0.606 | 13 | 0.506 | 17 | 73.50 | 5 |
| G20 | 1.330 | 19 | 0.332 | 15 | 0.055 | 15 | 1.770 | 19 | 0.554 | 19 | 3.930 | 19 | 0.759 | 18 | 0.575 | 19 | 51.00 | 12 |

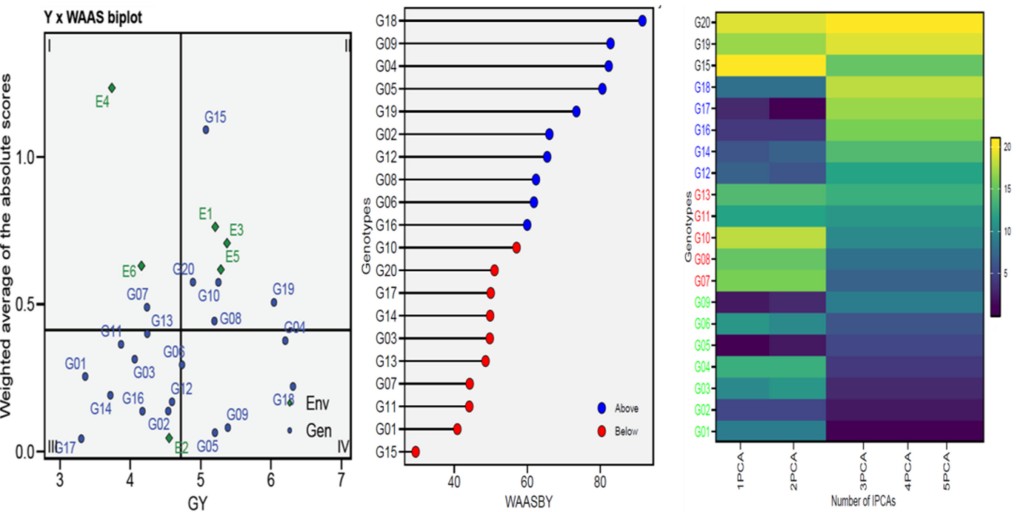

**Figure 3** WAAS analysis (WAAS biplot, WAAS and heatmap) for 20 wheat genotypes across six environments.

## WAAS biplot

To gain a more thorough and improved yield characterization (genotypes/environment), the WAASB analyses were utilized in selecting genotypes based on performance and stability (Fig. 3). Sector-I contains unstable genotypes with significant contributions to GEI and high distinction capacity, Sector-II contains unstable but highly productive genotypes where environments significantly influence GEI, Sector-III contains genotypes adopted on a larger scale with lower performance than average, indicating stable genotype performance across environments due to reduced WAASB values and Sector-IV contains genotypes with high performance and stability. For this, the G04, G05, G06, G09, and G18 genotypes were chosen for GY as perfect genotypes (Fig. 3). The genotype ranking (WAASBY) based on the weights of the stability (WAASB) and mean performance (Y) considering weights of 50 and 50 for GY, for the mean performance trait and WAASB (Fig. 3). Building on the number of IPCAs used in the WAASB assessment, the heatmap was used to show the genotype ranking of stable individuals (Fig. 3). The genotype's relative ranking is demonstrated by the color (intensity or hue), where higher ranks are represented by darker hues and lower rankings by lighter hues. Three IPCAs for traits were particularly noticeable, and the genotype ranking was modified by the IPCAs utilized in the WAASB assessment. Using genotype colors, it is easy to identify the groups with the same performance levels and stability (Fig. 3). The genotypes G01, G02, G03, G04, G05, G06, and G09 showed the lowest WAASB values (so were more stable), genotypes gathered in the same cluster (based on one or more IPCAs).

## Heat tolerance indices in GY trait
### Genetic (rg) and phenotypic (rp) correlations between GY and tolerance indices

The G10, G20, and G07 genotypes were less lost under heat stress, while the G15, G13, and G14 genotypes were greater (Table S3). The (rp) and (rg) values between $GY_{oc}$ and
$GY_{tsc}$ conditions, and with heat-tolerant indices for three seasons were computed to identify which approach would be best suited. The (rp) and (rg) values were positively significant between $GY_{oc}$ and $GY_{tsc}$, indicating that they can be used to recognize the best-performance genotypes (Table 5). Stress susceptibility index (SSI), RDI, percent yield reduction (PYR), and relative stress change (RSC) had a significant negative correlation with $GY_{tsc}$ but a positive correlation with $GY_{oc}$; so, these indices can be beneficial in selection for improving yields in non-stressed settings, but they may not be ideal for under more stressful environments with both (rp) and (rg) correlations. The stress tolerance index (STI), $STI_m$, yield index (YI), mean productivity (MP), geometric mean productivity (GMP), harmonic mean (HM), mean relative performance (MRP), relative heat index (RHI), stress-weighted performance index (SWPI), and stress-non-stress production index (SNPI) indices showed significant positive associations ($p \leq 0.01$) with Yp and Ys both (rp) and (rg) correlations, except for the SNPI index with rg, so they can be leveraged in detect genotypes that high-yield with Yp and Ys (Table 5). Some indices indicated a complete positive correlation ($r = 1.00$) for both (rp) and (rg), which is evidence of the collinearity of these indices, such as $GY_{tsc}$ with YI, TOL with stress susceptibility percentage index (SSPI), STI with $STI_m$ and REI, $STI_m$ with REI, SSI with PYR and RSC, MP and MRP, and PYR with RSC (Table 5).

### Variance components of indices traits

The LRT exhibited highly significant ($p < 0.001$) for all indices for both GEN and GEN: ENV, except for the SNPI index (Table 6). The variance components exhibited great variation between indices, the genotypic variance exhibited the highest value for the MP index and the lowest value for the SNPI index. The GEN: ENV exhibited the highest value for the RDI index and the lowest value for the SNPI index, and the residual exhibited the highest value for the SNPI index and the lowest value for the YI index. $h^2_{ems}$ showed mixed heritability values, in which most indices were more than 0.60, except for the RDI index (0.53), and the SNPI index is very low (0.18). The $h^2 mg$ exhibited more value compared to $h^2_{ems}$ for all indices. The accuracy exhibited a high value for all indices (>81.00%). The coefficient of variation (CVs) (g/r) ratio was greater than 1, except for the SNPI index. The $r_{gen:env}$ showed high values for all indices, which shows that the genotypic effect plays a major role in their inheritance, except for $GY_{oc}$ and SNPI indices.

### Factor identifying and selection of heat-tolerant genotypes

Principal component analysis (PCA) stated that the first two components (eigenvalue >1) illustrated 94.30% (before removing) and 87.80% (after removing) collinear variables of the cumulative variation among the 20 and 7 studied variables, respectively (Table 7). Before removing, FA illustrated that ten variables $GY_{oc}$, $GY_{tsc}$, STI, $STI_m$, YI, MP, GMP, HM, MRP, and REI were settling in FA1; and the remaining ten variables stress tolerance (TOL), SSPI, SSI, yield stability index (YSI), RDI, PYR, SWP, RSC, ATI and SNPI were settling in FA2. After removing variables, FA illustrated that four variables YSI, PYR, RSC, and SNPI were settling in FA1, and three variables STI, $STI_m$ and GMP were settling in FA2. The MGIDI index was used to identify the ideotype heat-tolerant after and before

**Table 5 Phenotypic (upper diagonal) and genotypic (below diagonal) correlations for GY and eighteen tolerance indices ($n = 180$).**

| | GY$_{oc}$ | GY$_{tsc}$ | TOL | STI | STI$_m$ | SSPI | SSI | YI | YSI | RDI | MP | GMP | HM | MRP | REI | PYR | SWP | RDC | ATI | SNPI |
|---|---|---|---|---|---|---|---|---|---|---|---|---|---|---|---|---|---|---|---|---|
| GY$_{oc}$ | 1.000 | 0.759 | 0.490 | 0.920 | 0.920 | 0.490 | **0.167** | 0.757 | **−0.170** | 0.234 | 0.945 | 0.629 | 0.897 | 0.936 | 0.919 | **0.170** | 0.456 | **0.170** | 0.716 | 0.216 |
| GY$_{tsc}$ | 0.757 | 1.000 | −0.195 | 0.944 | 0.944 | −0.196 | −0.509 | 1.000 | 0.507 | −0.445 | 0.930 | 0.619 | 0.968 | 0.940 | 0.945 | −0.507 | 0.924 | −0.507 | 0.094 | 0.706 |
| TOL | 0.497 | **−0.191** | 1.000 | 0.122 | 0.122 | 1.000 | 0.932 | −0.198 | −0.933 | 0.947 | **0.179** | 0.119 | 0.056 | **0.152** | 0.121 | 0.933 | −0.550 | 0.933 | 0.952 | −0.619 |
| STI | 0.920 | 0.943 | 0.129 | 1.000 | 1.000 | 0.122 | −0.207 | 0.943 | 0.204 | **−0.151** | 0.993 | 0.648 | 0.994 | 0.994 | 1.000 | −0.204 | 0.748 | −0.204 | 0.403 | 0.482 |
| STI$_m$ | 0.920 | 0.943 | 0.129 | 1.000 | 1.000 | 0.122 | −0.207 | 0.943 | 0.204 | **−0.151** | 0.993 | 0.648 | 0.994 | 0.994 | 1.000 | −0.204 | 0.748 | −0.204 | 0.403 | 0.482 |
| SSPI | 0.497 | **−0.191** | 1.000 | 0.128 | 0.128 | 1.000 | 0.933 | −0.198 | −0.934 | 0.947 | 0.179 | 0.118 | 0.055 | 0.152 | 0.120 | 0.934 | −0.550 | 0.934 | 0.952 | −0.619 |
| SSI | **0.174** | −0.505 | 0.932 | −0.201 | −0.201 | 0.932 | 1.000 | −0.511 | −1.000 | 0.968 | **−0.161** | −0.145 | −0.278 | −0.188 | −0.209 | 1.000 | −0.799 | 1.000 | 0.789 | −0.789 |
| YI | 0.755 | 1.000 | −0.193 | 0.943 | 0.943 | −0.193 | −0.507 | 1.000 | 0.509 | −0.447 | 0.929 | 0.618 | 0.967 | 0.939 | 0.944 | −0.509 | 0.925 | −0.509 | 0.091 | 0.707 |
| YSI | **−0.177** | 0.502 | −0.933 | 0.197 | 0.197 | −0.933 | −1.000 | 0.504 | 1.000 | −0.968 | **0.158** | 0.142 | 0.275 | **0.185** | 0.206 | −1.000 | 0.797 | −1.000 | −0.791 | 0.790 |
| RDI | 0.242 | −0.439 | 0.947 | −0.143 | −0.143 | 0.947 | 0.967 | −0.441 | −0.967 | 1.000 | −0.091 | −0.115 | −0.218 | −0.118 | **−0.152** | 0.968 | −0.738 | 0.968 | 0.810 | −0.654 |
| MP | 0.945 | 0.929 | **0.186** | 0.993 | 0.993 | **0.186** | −0.154 | 0.928 | **0.151** | −0.082 | 1.000 | 0.665 | 0.992 | 1.000 | 0.993 | **−0.158** | 0.721 | **−0.158** | 0.451 | 0.476 |
| GMP | 0.630 | 0.619 | **0.126** | 0.649 | 0.649 | 0.125 | −0.138 | 0.618 | 0.135 | −0.107 | 0.667 | 1.000 | 0.663 | 0.665 | 0.648 | −0.142 | 0.495 | −0.142 | 0.340 | 0.365 |
| HM | 0.896 | 0.967 | 0.062 | 0.993 | 0.993 | 0.062 | −0.271 | 0.967 | 0.268 | **−0.210** | 0.991 | 0.664 | 1.000 | 0.995 | 0.994 | −0.275 | 0.798 | −0.275 | 0.340 | 0.542 |
| MRP | 0.936 | 0.939 | **0.159** | 0.994 | 0.994 | **0.159** | −0.181 | 0.938 | **0.177** | −0.109 | 1.000 | 0.666 | 0.994 | 1.000 | 0.994 | **−0.185** | 0.739 | **−0.185** | 0.427 | 0.495 |
| REI | 0.919 | 0.944 | 0.127 | 1.000 | 1.000 | 0.127 | −0.202 | 0.943 | 0.199 | −0.144 | 0.993 | 0.649 | 0.994 | 0.994 | 1.000 | −0.206 | 0.749 | −0.206 | 0.401 | 0.483 |
| PYR | **0.177** | −0.502 | 0.933 | −0.197 | −0.197 | 0.933 | 1.000 | −0.504 | −1.000 | 0.967 | **−0.151** | −0.135 | −0.268 | **−0.177** | −0.199 | 1.000 | −0.797 | 1.000 | 0.791 | −0.790 |
| SWP | 0.451 | 0.923 | −0.548 | 0.745 | 0.745 | −0.548 | −0.798 | 0.924 | 0.796 | −0.736 | 0.717 | 0.492 | 0.795 | 0.736 | 0.746 | −0.796 | 1.000 | −0.797 | −0.286 | 0.851 |
| RDC | **0.177** | −0.502 | 0.933 | **−0.197** | **−0.197** | 0.933 | 1.000 | −0.504 | −1.000 | 0.967 | **−0.151** | −0.135 | −0.268 | **−0.177** | −0.199 | 1.000 | −0.796 | 1.000 | 0.791 | −0.790 |
| ATI | 0.719 | 0.096 | 0.953 | 0.406 | 0.406 | 0.953 | 0.789 | 0.093 | −0.791 | 0.812 | 0.455 | 0.344 | 0.343 | 0.431 | 0.404 | 0.791 | −0.287 | 0.791 | 1.000 | −0.466 |
| SNPI | **0.177** | 0.579 | −0.502 | 0.396 | 0.396 | −0.502 | −0.643 | 0.580 | 0.644 | −0.532 | 0.390 | 0.305 | 0.444 | 0.406 | 0.396 | −0.644 | 0.697 | −0.644 | −0.378 | 1.000 |

**Notes.**

Values in bold are significant at 0.05, underlined values are insignificant, and the remaining values are significant at 0.01.

**Table 6 Deviance analysis, estimated variance components and genetic parameters for GY and eighteen tolerance indices of 20 wheat genotypes.**

| Genetic parameters | | $GY_{oc}$ | $GY_{tsc}$ | TOL | STI | $STI_m$ | SSPI | $SSI_m$ | YI | YSI | RDI |
|---|---|---|---|---|---|---|---|---|---|---|---|
| GEN | $x^2$ | 80.212 | 54.661 | 26.275 | 70.136 | 70.136 | 26.390 | 28.067 | 54.692 | 27.488 | 16.713 |
| | $p$-value | $3.36 \times 10^{-19}$ | $1.43 \times 10^{-13}$ | $2.96 \times 10^{-7}$ | $5.54 \times 10^{-17}$ | $5.54 \times 10^{-17}$ | $2.79 \times 10^{-7}$ | $1.17 \times 10^{-7}$ | $1.41 \times 10^{-13}$ | $1.58 \times 10^{-7}$ | $4.35 \times 10^{-5}$ |
| GEN:ENV | $x^2$ | 15.368 | 189.257 | 68.862 | 81.831 | 81.831 | 68.462 | 79.844 | 191.044 | 82.041 | 132.613 |
| | $p$-value | $8.85 \times 10^{-5}$ | $4.62 \times 10^{-43}$ | $1.06 \times 10^{-16}$ | $1.48 \times 10^{-19}$ | $1.48 \times 10^{-19}$ | $1.29 \times 10^{-16}$ | $4.05 \times 10^{-19}$ | $1.88 \times 10^{-43}$ | $1.33 \times 10^{-19}$ | $1.10 \times 10^{-30}$ |
| GEN | | 0.964 | 0.731 | 0.377 | 0.096 | 15326.83 | 15326.83 | 25086.99 | 0.036 | 0.010 | 0.05 |
| GEN:ENV | | 0.046 | 0.119 | 0.158 | 0.009 | 1412.55 | 1412.55 | 10470.11 | 0.006 | 0.004 | 0.037 |
| Residual | | 0.085 | 0.013 | 0.075 | 0.003 | 542.98 | 542.98 | 4966.57 | 0.001 | 0.002 | 0.007 |
| Phenotypic variance | | 1.09 | 0.863 | 0.609 | 0.108 | 17282.36 | 40523.67 | 0.864 | 0.043 | 0.015 | 0.094 |
| $h^2_{ems}$ | | 0.881 | 0.848 | 0.618 | 0.887 | 0.887 | 0.619 | 0.644 | 0.848 | 0.639 | 0.528 |
| $R^2_{gen:env}$ | | 0.042 | 0.138 | 0.26 | 0.082 | 0.082 | 0.258 | 0.255 | 0.138 | 0.261 | 0.394 |
| $h^2_{pm}$ | | 0.975 | 0.947 | 0.861 | 0.967 | 0.967 | 0.861 | 0.87 | 0.947 | 0.867 | 0.79 |
| Accuracy | | 0.987 | 0.973 | 0.928 | 0.983 | 0.983 | 0.928 | 0.933 | 0.973 | 0.931 | 0.889 |
| $r_{gen:env}$ | | 0.354 | 0.904 | 0.68 | 0.722 | 0.722 | 0.678 | 0.716 | 0.906 | 0.723 | 0.835 |
| $CV_{gen}$ | | 18.6 | 20.606 | 53.856 | 36.567 | 36.567 | 53.853 | 46.927 | 20.625 | 12.388 | 14.739 |
| $CV_{res}$ | | 5.49 | 2.704 | 23.957 | 6.883 | 6.883 | 23.961 | 18.591 | 2.681 | 4.898 | 5.657 |
| CV ratio | | 3.38 | 7.621 | 2.248 | 5.313 | 5.313 | 2.247 | 2.524 | 7.693 | 2.529 | 2.606 |
| Genetic parameters | | MP | GMP | HM | MRP | REI | PYR | SWP | RDC | ATI | SNPI |
| GEN | $x^2$ | 83.424 | 57.655 | 64.996 | 81.818 | 70.539 | 27.488 | 38.274 | 27.488 | 28.794 | 9.205 |
| | $p$-value | $6.62 \times 10^{-20}$ | $3.12 \times 10^{-14}$ | $7.50 \times 10^{-16}$ | $1.49 \times 10^{-19}$ | $4.51 \times 10^{-17}$ | $1.58 \times 10^{-7}$ | $6.15 \times 10^{-10}$ | $1.58 \times 10^{-7}$ | $8.05 \times 10^{-8}$ | $24 \times 10^{-2}$ |
| GEN:ENV | $x^2$ | 47.787 | 84.701 | 108.290 | 56.016 | 81.965 | 82.041 | 161.146 | 82.041 | 54.712 | 0.000 |
| | $p$-value | $4.75 \times 10^{-12}$ | $3.47 \times 10^{-20}$ | $2.32 \times 10^{-25}$ | 0.781 | $1.39 \times 10^{-19}$ | $1.33 \times 10^{-19}$ | $6.36 \times 10^{-37}$ | $1.33 \times 10^{-19}$ | $1.40 \times 10^{-13}$ | 1.00 |
| GEN | | 0.783 | 0.483 | 0.783 | 0.129 | 0.126 | 96.03 | 0.068 | 96.03 | 0.0181 | 60.966 |
| GEN:ENV | | 0.043 | 0.059 | 0.082 | 0.008 | 0.011 | 39.183 | 0.019 | 39.183 | 0.0065 | 0 |
| Residual | | 0.030 | 0.022 | 0.022 | 0.005 | 0.004 | 15.015 | 0.003 | 15.015 | 0.004 | 273.278 |
| Phenotypic variance | | 0.826 | 0.51 | 0.836 | 0.142 | 0.142 | 150.227 | 0.09 | 150.227 | 0.029 | 334.244 |
| $h^2_{ems}$ | | 0.912 | 0.843 | 0.876 | 0.911 | 0.888 | 0.639 | 0.755 | 0.639 | 0.633 | 0.182 |
| $R^2_{gen:env}$ | | 0.052 | 0.115 | 0.098 | 0.056 | 0.081 | 0.261 | 0.214 | 0.261 | 0.229 | 0 |
| $h^2_{pm}$ | | 0.977 | 0.952 | 0.961 | 0.976 | 0.967 | 0.867 | 0.91 | 0.867 | 0.873 | 0.668 |
| Accuracy | | 0.988 | 0.975 | 0.98 | 0.988 | 0.983 | 0.931 | 0.954 | 0.931 | 0.935 | 0.817 |
| $r_{gen:env}$ | | 0.59 | 0.731 | 0.79 | 0.629 | 0.723 | 0.723 | 0.875 | 0.723 | 0.623 | 0 |
| $CV_{gen}$ | | 18.389 | 14.148 | 18.561 | 18.451 | 36.565 | 46.898 | 14.454 | 46.898 | 58.575 | 61.628 |
| $CV_{res}$ | | 3.663 | 3.169 | 3.201 | 3.518 | 6.835 | 18.544 | 2.909 | 18.544 | 27.404 | 130.478 |
| CV ratio | | 5.02 | 4.465 | 5.799 | 5.244 | 5.35 | 2.529 | 4.969 | 2.529 | 2.137 | 0.472 |
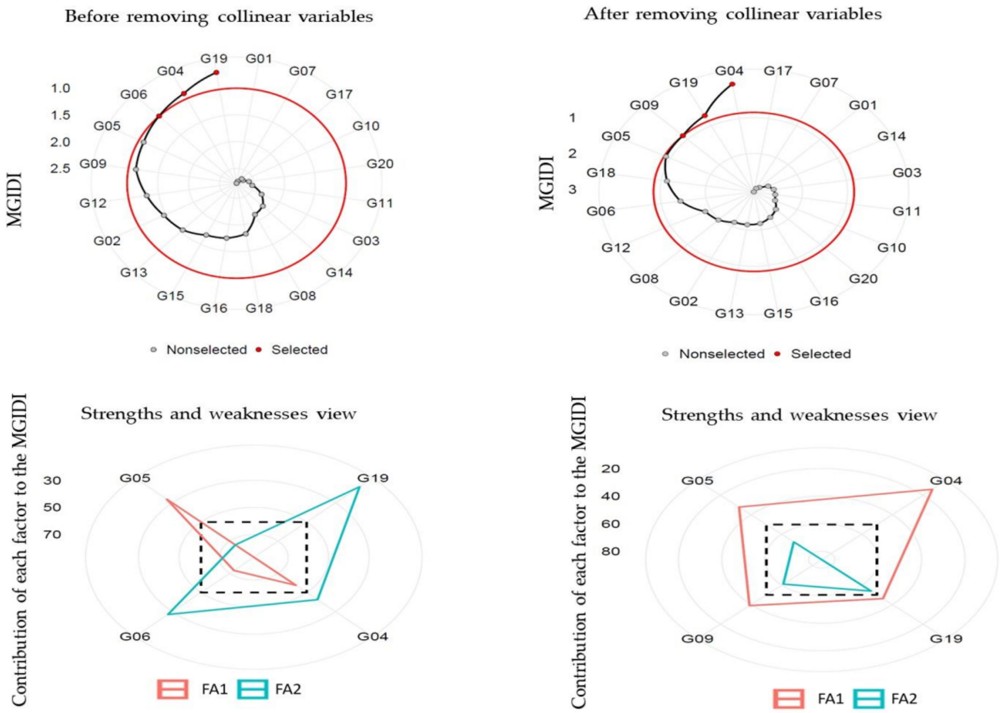

**Figure 4 Genotype ranking for the MGIDI and strengths and weaknesses view of the selected genotypes.**

removing collinear. The selection gains (MGIDI index) before removing revealed that 13 out of 20 variables were desired gains, and four out of seven after removing collinear variables. The results illustrated that MGIDI showed higher total gains of 345.96 and 106.54 for variables that increased and −5.78 and −1.757 for variables that decreased before and after removing variables, respectively (Table 8). The abiotic tolerance index (ATI), REI, and SSPI illustrated the highest genetic gains (43.90%, 33.00%, and 26.10%, respectively) before removing variables, but after removing variables were $STI_m$ (40.100) and GMP (19.50). The MGIDI index of the original population (Xo) before and after removing variables varied from 0.229 and 0.846 (the lowest one), for the ATI and STI to 339.00 (the highest one) for the $STI_m$, respectively (Table 8). The genotypes selected using the MGIDI were G04, G05, G06, and G19 before removing variables and they were G04, G05, G09, and G19 after removing variables (Fig. 4). The G05 was very close to the cutting point before and after removing variables. The strengths and weaknesses illustrated that before removing variables, FA1 had the highest contribution for G04, G06 and G19. FA2 had the highest contribution for G05 (Fig. 4). But after removing variables, FA2 had the highest contribution for the four genotypes, while FA1 didn't have any contributions.

## DISCUSSION

Thermal stress is one of the biggest environmental stresses negatively impacting wheat yields across wheat-growing countries. Breeding programs focus on enhancing the genetics

**Table 7 PCA and FA with factorial loadings obtained using varimax rotation and communalities resulted.**

| | All traits before removing colinear variables | | | | Selected traits after removing colinear variables | | | |
|---|---|---|---|---|---|---|---|---|
| | PCA | | | | PCA | | | |
| PCA | PC1 | PC2 | PC3 | PC4 | PC1 | PC2 | PC3 | PC4 |
| Eigenvalues | 10.90 | 7.94 | 0.55 | 0.493 | 3.92 | 2.19 | 0.46 | 0.43 |
| Variance (%) | 54.60 | 39.70 | 2.76 | 2.47 | 56.00 | 31.30 | 6.50 | 6.17 |
| Cumul (%)[*] | 54.60 | 94.30 | 97.10 | 99.6 | 56.00 | 87.30 | 93.80 | 100.00 |
| | FA | | | | FA | | | |
| Variable | FA1 | FA2 | Comm[#] | Uniqu[$] | FA1 | FA2 | Comm[#] | Uniqu[$] |
| GYoc | **0.938** | −0.343 | 0.996 | 0.004 | | | | |
| GYtsc | **0.933** | 0.352 | 0.995 | 0.005 | | | | |
| TOL | 0.169 | **−0.982** | 0.994 | 0.006 | | | | |
| STI | **0.993** | 0.034 | 0.988 | 0.012 | 0.133 | **−0.958** | 0.935 | 0.065 |
| STI$_m$ | **0.993** | 0.034 | 0.988 | 0.012 | −0.133 | **0.958** | 0.935 | 0.065 |
| SSPI | −0.169 | **0.982** | 0.994 | 0.006 | | | | |
| SSI | 0.172 | **0.981** | 0.992 | 0.008 | | | | |
| YI | **−0.932** | −0.355 | 0.995 | 0.005 | | | | |
| YSI | −0.168 | **−0.982** | 0.992 | 0.008 | **−0.989** | 0.061 | 0.981 | 0.019 |
| RDI | 0.103 | **0.970** | 0.951 | 0.049 | | | | |
| MP | **−0.998** | 0.018 | 0.996 | 0.004 | | | | |
| GMP | **−0.706** | −0.003 | 0.498 | 0.502 | −0.074 | **0.810** | 0.662 | 0.338 |
| HM | **−0.992** | −0.104 | 0.995 | 0.005 | | | | |
| MRP | **−0.998** | −0.010 | 0.996 | 0.004 | | | | |
| REI | **−0.993** | −0.035 | 0.988 | 0.012 | | | | |
| PYR | 0.168 | **0.982** | 0.992 | 0.008 | **0.989** | −0.061 | 0.981 | 0.019 |
| SWP | −0.727 | **−0.684** | 0.996 | 0.004 | | | | |
| RDC | 0.168 | **0.982** | 0.992 | 0.008 | **0.989** | −0.061 | 0.981 | 0.019 |
| ATI | −0.443 | **0.889** | 0.987 | 0.013 | | | | |
| SNPI | −0.409 | **−0.612** | 0.542 | 0.458 | **−0.713** | 0.360 | 0.638 | 0.362 |

**Notes.**

Values in bold refer to critical variable on FA.

[#]Communality.

[$]Uniquenesses.

[*]Cumulative variance (%).

of increased tolerance to heat stress for better yields, they are essential to maintaining food security and sustainability in the face of shifting environmental conditions and global problems because they constantly innovate and modify breeding tactics. Their efforts result in developing new and enhanced genotypes that fulfill the demands of an expanding population and strengthen the agricultural system (*Lamba et al., 2023*; *Motawei, Kamara & Rehan, 2025*). This study uses twenty wheat genotypes to evaluate their GY under two conditions (optimum and thermal stress) for three seasons and found that thermal stress influenced GY negatively (Fig. 1). The wheat genotype's performance variances under optimum conditions were higher compared to the thermal stress and the cause may be the accumulation of low biomass (due to the negative impact of growth traits like spike length,

**Table 8  Predicted genetic gains for the indexes MGIDI for all variables and selected variables before and after removing colinear variables.**

| | | All variables before removing colinear variables | | | | | | | | | Selected variables after removing colinear variables | | | | | | |
|---|---|---|---|---|---|---|---|---|---|---|---|---|---|---|---|---|---|
| | Var | Xo | Xs | SD | SDperc | SG | MGIDI | sense | FA | Var | Xo | Xs | SD | SDperc | SG | MGIDI | sense |
| FA1 | $GY_{oc}$ | 5.29 | 6.32 | 1.03 | 19.4 | 0.986 | 18.60 | decrease | FA1 | YSI | 0.791 | 0.782 | −0.009 | −1.200 | −0.007 | −0.905 | increase |
| FA1 | $GY_{tsc}$ | 4.15 | 4.74 | 0.592 | 14.3 | 0.538 | 13.00 | decrease | FA1 | PYR | 20.90 | 21.800 | 0.947 | 4.530 | 0.716 | 3.420 | increase |
| FA1 | STI | 0.846 | 1.14 | 0.291 | 34.3 | 0.278 | 32.90 | decrease | FA1 | RDC | 20.90 | 21.800 | 0.947 | 4.530 | 0.716 | 3.420 | increase |
| FA1 | $STI_m$ | 339 | 455 | 116 | 34.3 | 111 | 32.90 | decrease | FA1 | SNPI | 12.70 | 12.400 | −0.270 | −2.130 | −0.108 | −0.852 | increase |
| FA1 | YI | 0.923 | 1.05 | 0.132 | 14.3 | 0.121 | 13.10 | increase | FA2 | STI | 0.846 | 1.200 | 0.355 | 41.900 | 0.340 | 40.100 | decrease |
| FA1 | MP | 4.72 | 5.54 | 0.817 | 17.3 | 0.789 | 16.70 | increase | FA2 | $STI_m$ | 339.0 | 480.00 | 142.00 | 41.900 | 136.00 | 40.100 | increase |
| FA1 | GMP | 4.64 | 5.51 | 0.871 | 18.8 | 0.814 | 17.60 | increase | FA2 | GMP | 4.640 | 5.600 | 0.967 | 20.900 | 0.904 | 19.500 | increase |
| FA1 | HM | 4.61 | 5.39 | 0.775 | 16.8 | 0.733 | 15.90 | increase | | | | | | | | | |
| FA1 | MRP | 1.95 | 2.28 | 0.333 | 17.1 | 0.323 | 16.60 | increase | | | | | | | | | |
| FA1 | REI | 0.971 | 1.30 | 0.333 | 34.3 | 0.32 | 33.00 | increase | | | | | | | | | |
| FA1 | SWP | 1.80 | 1.89 | 0.085 | 4.72 | 0.071 | 3.92 | increase | | | | | | | | | |
| FA2 | TOL | 1.14 | 1.53 | 0.389 | 34.1 | 0.297 | 26.00 | decrease | | | | | | | | | |
| FA2 | SSPI | 294 | 394 | 100 | 34.1 | 76.7 | 26.10 | increase | | | | | | | | | |
| FA2 | SSI | 1.59 | 1.82 | 0.231 | 14.6 | 0.178 | 11.20 | increase | | | | | | | | | |
| FA2 | YSI | 0.791 | 0.76 | −0.031 | −3.91 | −0.023 | −2.95 | increase | | | | | | | | | |
| FA2 | RDI | 1.51 | 1.56 | 0.045 | 3.00 | 0.032 | 2.14 | increase | | | | | | | | | |
| FA2 | PYR | 20.9 | 24 | 3.09 | 14.8 | 2.33 | 11.20 | increase | | | | | | | | | |
| FA2 | RDC | 20.9 | 24 | 3.09 | 14.8 | 2.33 | 11.20 | increase | | | | | | | | | |
| FA2 | ATI | 0.229 | 0.356 | 0.127 | 55.4 | 0.101 | 43.90 | increase | | | | | | | | | |
| FA2 | SNPI | 12.7 | 11.8 | −0.895 | −7.07 | −0.358 | −2.83 | increase | | | | | | | | | |
| Total (increase) | | | | 80.53 | | | 345.96 | | | | | | | | | 106.540 | |
| Total (decrease) | | | | −1.28 | | | −5.78 | | | | | | | | | −1.757 | |

grains/spike, and thousand-kernel weight) in tandem with high-temperature in a month before the end of the growing season (*Arif et al., 2025*; *Farhad et al., 2023*; *Fu et al., 2023*).

The joint ANOVA and AMMI model analyses indicated variable genotype performance with thermal stress indicating the presence of genetic variations in the genotypes used for heat tolerance (Table 2). The GEN: ENV interaction was significant, adversely affecting selection efficiency due to varying genotype rankings (*Al-Ashkar et al., 2022*; *Erdemcı, 2018*; *Sampaio Filho et al., 2023*). To mitigate bias and increase confidence in selection gains, multi-environment trials (METs) should be utilized. METs provide valuable insights for breeders aiming to enhance resilience in wheat production. This study's the challenge of choosing wheat genotypes that successfully strike a compromise between stability and excellent performance. It employs innovative statistics to analyze genetic parameters, enabling the identification of genotypes that are resilient to the negative effects of thermal stress (*Al-Ashkar, 2024*; *Al-Ashkar et al., 2022*; *Olivoto & Nardino, 2021*; *Pour-Aboughadareh et al., 2021*; *Pour-Aboughadareh et al., 2019*). The pooled ANOVA as per Eberhart and Russell (*Eberhart & Russell, 1966*), demonstrated significant distinctions for all model effects, indicating that the genotype performance varied by ENV. Many scholars found the same outcome (*Al-Ashkar et al., 2022*; *Al-Ashkar et al., 2023a*; *Pour-Aboughadareh et al., 2021*). In this study, the two genotypes G05 and G09 had bi values close to 1 indicating that they are more stable under every six ENVs (Table 3). Genotypes G04, G06, G12, G13, G15 and G19 were observed to be stable in optimum conditions (ENV1, ENV3 and ENV5) environments, and genotypes G08, G10, and G18 were more resilience to thermal stress (ENV2, ENV4 and ENV6) environments (*Al-Ashkar et al., 2023b*; *Eberhart & Russell, 1966*; *Gupta et al., 2022*; *Suresh & Munjal, 2020*). As to stability indices, differing ranking were expressed, but some are compatible with each other (Table 4).

The Annicchiarico method pointed out that there was consensus acceptable in ranking between the favorable and unfavorable environment in tandem with general analysis, this was consensus acceptable with results Eberhart and Russell in Table 3. Our findings indicate ASI, ASV, MASI, MASV, and WAAS were matched in ranking genotypes, DA with FA, and DZ with EV were matched. Biplots-AMMI has the featured of taking all IPCA axes, enabling GEN: ENV not retained in the first IPCA axis for inclusion in the ranking of genotypes (*Al-Ashkar, 2024*; *Olivoto et al., 2019a*). In this study, AMMI revealed that the sum of squares for the environment was divided into the first two significant components of 99.00% (Table 2). The AMMI1 biplot illustrated the GEN: ENV, which makes it clear that when it is far from its origin and has a longer vector, it exhibits higher interaction, as seen in the ENV4 (*Ahmed et al., 2024*; *Al-Ashkar, 2024*; *Ebdon & Gauch, 2002*; *Mebratu et al., 2019*; *Popovic et al., 2020*; *Singamsetti et al., 2021*). Conversely, the ENVs that are close to their origin and have shorter vectors, such as ENV2, indicate less interaction. The angles among the vectors of optimal conditions (ENV1, ENV3, and ENV5) were less than 90°, demonstrating a positive correlation between them. Similarly, the angles between the vectors of thermal stress conditions (ENV2, ENV4, and ENV6) also show a positive correlation. It means that GEN: ENV effects tend to be independent and within the same range when applied under similar circumstances (Fig. 2). The GGE biplot polygon

has been used to establish his identity of the most desirable genotypes that exhibit high discriminativeness and representativeness, and are located in the upper right quarter in the polygon (G04, G14, G15, G16, and G19). A vertical projection from the GEN to the ENV vector shows the GEN: ENV volume with the ENVs (*Al-Ashkar, 2024*; *Al-Ashkar et al., 2022*; *Habib et al., 2024*; *Singamsetti et al., 2021*). Thus, the genotypes that might be viewed (G03, G05, G06, G08, G09, G10, G14, G15, and G20) are deemed unstable with the six ENVs used (Fig. 2). The adaption map showed that G05, G09, and G17 were better suited and exhibited similar performance in the ENVs (*Al-Ashkar et al., 2023b*; *Habib et al., 2024*; *Olivoto et al., 2019a*). Although their performance varies from ENV to other, the G15 performed best. The WAASB employs a unique method for selecting genotypes that exhibit both high performance and stability by considering all IPCAs. This approach successfully illustrates the GEN: ENV for its combination of AMMI and BLUP models (*Ahmed et al., 2024*; *Al-Ashkar et al., 2023b*; *Olivoto et al., 2019a*; *Olivoto et al., 2019b*; *Pour-Aboughadareh et al., 2021*). Depending on the WAAS and GY values, a WAAS biplot determined genotypes of best performance and stability, which are located in the bottom right quartile (Sector IV) as shown in Fig. 3. This method takes all IPCAs into account and reduces redundancy, making it a promising approach for discovering high-performing and stable genotypes in future research, and will facilitate the process of recommending ideotype cultivars (*Ahmed et al., 2024*; *Al-Ashkar, 2024*; *Olivoto et al., 2019b*). The heatmap demonstrated genotypes ranking by color (intensity or hue), where higher ranks are represented by darker hues and lower ranks by lighter hues (*Ahmed et al., 2024*; *Al-Ashkar, 2024*). The genotypes G01, G02, G03, G04, G05, G06, and G09 showed the lowest WAASB values (so they were more stable and performed well), and were grouped in the same cluster (based on one or more IPCAs). This is crucial in breeding programs, as breeders may be given a greater priority to high performance than stability or vice versa, therefore, Fig. 3 can assist breeders in making informed decisions about selecting genotypes that exhibit similar mean performance and stability (*Olivoto et al., 2019a*). In addition to its prospective breeding importance, as a genetic source in constant development programs aimed at creating high-performance, thermal stress-tolerant new varieties.

The AMMI-ANOVA results indicated significant differences in GEN: ENV and genotype performance varied under optimal and thermal stress conditions, demonstrating that each genotype reacted differently in the two conditions for GY. For this reason, plant breeders employ various methods to select high-yielding genotypes in thermal stress conditions, known as the stress-tolerance index (STI) or ''selection indices''. These indices are widely used in research to identify genotypes capable of assuming thermal stress (*Kumar et al., 2021*; *Kumar et al., 2023*; *Lamba et al., 2023*; *Poudel, Poudel & Puri, 2021*). Higher values in TOL, RDI, SSI, ATI, and SSPI hint at more sensitive genotypes, but the lower values tolerant genotypes. These indices are maligned in that they cannot differentiate between the genotype's high yield (*Al-Ashkar, 2024*; *Lamba et al., 2023*). The G10, G20, and G07 genotypes were less lost under thermal stress, and the distinctions between their values under (optimal and thermal stress) conditions were minimal, while the G15, G13, and G14 genotypes were more lost (Table S3). Many scientists, *Shabani et al. (2018)*, *Kamrani, Hoseini & Ebadollahi (2017)*, *Lamba et al. (2023)* and *Al-Ashkar (2024)* reported STI, MP,

and GMP indices are the most appropriate to choose the more tolerant genotypes and more productive through higher values of indices such as G04, G10, G18 and G19 genotypes. So, we carried out genetic (rg) and phenotypic (rp) correlation analyses between GY (under optimal condition (OC) and thermal stress condition (TSC)) and tolerance indices to achieve the most appropriate indices for thermal stress tolerance. $GY_{oc}$ and $GY_{tsc}$ both had a positive correlation, which facilitate the identification of high-performance genotypes based on $GY_{oc}$ and $GY_{tsc}$, allowing indirect selection for $GY_{tsc}$ through $GY_{oc}$. The TOL, SSPI, RDI, PYR, SSI, and RDC showed a negative with $GY_{tsc}$ but and a positive correlation with $GY_{oc,}$; thus, selection according to these indices will improve productivity with optimal conditions but lower it with thermal stress conditions (*vice versa*) (Table 5). Ten out of eighteen indices showed a positive and significant correlation with both $GY_{oc}$ and $GY_{tsc}$, which could be used to detect highly productive genotypes in both $GY_{oc}$ and $GY_{tsc}$ (*Al-Ashkar, 2024*; *Basavaraj et al., 2021*; *Kumar et al., 2023*; *Lamba et al., 2023*).

This study highlighted the importance of selection indices characterized by strong genetic stability using cutting-edge statistical techniques to better understand genetic factors and identify indices that are least influenced by the environment. The $\sigma^2_{gen}$ value (more than 52.81% from $\sigma^2_{phenotypic\ total}$) exceeded the $\sigma^2_{res}$ value (less than 13.85% from $\sigma^2_{phenotypic\ total}$) for selection indices (increasing heritability), indicating the right conditions to choose genotype during the various phases of the breeding program, except SNPI index, which very low for $\sigma^2_{gen}$ and very high for $\sigma^2_{res}$ (reducing heritability) (Table 6). The $r_{gen:env}$ showed high values for all indices, except for SNPI index. The high value indicates that the genotypic effect is predominant, while the interaction effect is simple; consequently, low values are undesirable for genotype selection (*Al-Ashkar, 2024*; *Al-Ashkar et al., 2023c*; *Olivoto et al., 2019b*). The $h^2_{ems}$ showed mixed heritability values and most indices were more than 0.60, which reflects a significant increase in genetic diversity (the accuracy degree of more than 0.81), except for the SNPI index. This high degree of accuracy suggests a strong ability to predict genetic worth (*Al-Ashkar et al., 2023b*; *Sampaio Filho et al., 2023*). The CV (g/r) ratio was greater than 1, indicating that genetic variation (CVg) exceeded residual variation (CVr) (*Al-Ashkar, 2024*; *Olivoto et al., 2019a*).

The MGIDI is one new statistical technique that assists in detecting a better genotype of a broad range of variables at a time (*Azam et al., 2020*; *Khyathi et al., 2025*; *Olivoto & Nardino, 2021*). The genotype selection process based on one variable is not preferred by plant breeders because could mislead interpretations of the results (*Al-Ashkar, 2024*; *Olivoto et al., 2019b*). Therefore, the MGIDI is beneficial in the genotype selection process based on a broad range of variables since it offers a selection process clear and intelligible (Table 7). The distance is computed for genotype-ideotype using a factor analysis (*Olivoto & Nardino, 2021*). Based on the variables under evaluation, the selection gains (MGIDI index to identify the ideotype heat-tolerant) before removing revealed that 13 out of 20 variables were desired gains, and four out of seven after removing collinear variables (Table 8). The most desirable or stable genotypes are believed to be G04, G05, G06, and G19 before removing variables and G04, G05, G09, and G19 after removing variables since the genotypes with lower MGIDI index values have better stability. In both situations, the G05 was present. A distinct and easy-to-understand selection process unique with

numerous practical applications to obtaining long-term genetic gain is the MGIDI index (*Al-Ashkar, 2024*; *Habib et al., 2024*; *Olivoto & Nardino, 2021*; *Salami et al., 2025*; *Sampaio Filho et al., 2023*). The proportion interpreted by every factor is another benefit of the MGIDI index "strengths and weaknesses view", a crucial graphical tool for determining the strengths and weaknesses of test hybrids in terms of "trait (group of traits) need to be improved" in subsequent hybridization programs to produce new recombination known as the ideotype (Fig. 4). For instance, future research could explore crossbreeding genotype G05 with G04, G06, or G19 to develop a novel recombinant ideotype combining all desired selection indices. The implementation of the MGIDI index makes it easier to provide recommendations for improved crop cultivars and allows for more informed strategic decision-making in stability evaluation studies by facilitating the minimization of redundant calculations (*Al-Ashkar et al., 2023b*; *Habib et al., 2024*; *Khyathi et al., 2025*; *Olivoto & Nardino, 2021*).

## CONCLUSIONS

This study demonstrates that integrating stability analysis (AMMI, WAASB) with multi-trait selection (MGIDI) provides an effective framework for identifying climate-resilient wheat genotypes. The approach successfully distinguished genotypes combining yield stability (G05, G09, G17) and high performance (G04, G05, G06, G09) under both optimal and thermal stress conditions. Notably, the strong concordance between statistical models and selection indices validates their combined use for stress-resilience breeding. This study establishes a reproducible selection protocol that prioritizes both agronomic performance and environmental stability—critical criteria for developing climate-ready wheat varieties. By bridging the gap between phenotypic stability and breeding objectives, this strategy offers a scalable solution for genotype selection in increasingly variable environments. Future efforts should focus on validating these genotypes across broader agro-ecological zones while incorporating genomic tools to accelerate selection.

### Funding

The authors received funding from the Ongoing Research Funding program, ORF-2025-298), King Saud University, Riyadh, Saudi Arabia. The funders had no role in study design, data collection and analysis, decision to publish, or preparation of the manuscript.

### Grant Disclosures

The following grant information was disclosed by the authors:
King Saud University, Riyadh, Saudi Arabia: ORF-2025-298.

### Competing Interests

The authors declare there are no competing interests.

## Author Contributions

- Abdelhalim Ghazy analyzed the data, prepared figures and/or tables, authored or reviewed drafts of the article, and approved the final draft.
- Walid Ben Romdhane analyzed the data, prepared figures and/or tables, authored or reviewed drafts of the article, and approved the final draft.
- Majed Alotaibi analyzed the data, authored or reviewed drafts of the article, and approved the final draft.
- Abdullah Al-Doss performed the experiments, authored or reviewed drafts of the article, and approved the final draft.
- Omar Dahrog performed the experiments, authored or reviewed drafts of the article, and approved the final draft.
- Nasser Al-Suhaibani analyzed the data, authored or reviewed drafts of the article, and approved the final draft.
- Abdullah Ibrahim performed the experiments, authored or reviewed drafts of the article, and approved the final draft.
- Adel M. Al-Saif analyzed the data, prepared figures and/or tables, authored or reviewed drafts of the article, and approved the final draft.
- Khalid A. Al-Gaadi performed the experiments, analyzed the data, authored or reviewed drafts of the article, and approved the final draft.
- Ahmed M. Zeyada performed the experiments, authored or reviewed drafts of the article, and approved the final draft.
- Khalid F. Almutairi performed the experiments, authored or reviewed drafts of the article, and approved the final draft.
- Ibrahim Al-Ashkar conceived and designed the experiments, performed the experiments, analyzed the data, prepared figures and/or tables, authored or reviewed drafts of the article, and approved the final draft.

## Data Availability

The raw measurements are available in the Supplemental Files.

## Supplemental Information

Supplemental information for this article can be found online at http://dx.doi.org/10.7717/peerj.20061#supplemental-information.

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
