# Peer review of "Selection of suitable wheat genotypes under thermal stress and complex genotype-environment interaction using stability analyses and selection indices"

_PeerJ, doi:10.7717/peerj.20061_

## Round 0.1 · original submission · Major Revisions

Dear Authors

The manuscript cannot be accepted for publication in its current form. It needs a major revision before publication. The authors are invited to revise the paper, considering all the suggestions made by the reviewers. Please note that the requested changes are required for publication.

With Thanks

Reviewer 1 ·

Basic reporting

Dear author
I have carefully reviewed the manuscript titled "Selection of Suitable Wheat Genotypes Under Thermal Stress and Complex Genotype-Environment Interaction Using Stability Analyses and Selection Indices" presents an important topic relevant to wheat breeding under climate change conditions. However, the manuscript, in its current form, has several significant shortcomings that need to be addressed before it can be considered for publication.
Major Concerns and Required Revisions:
Novelty and Justification:
The study lacks a clear statement of novelty. Given that extensive research has been conducted on wheat genotype selection under stress conditions, the authors must explicitly state what makes this study unique and how it advances existing knowledge.
The introduction should highlight specific gaps in the literature that this study aims to fill.
Introduction Needs More Depth:
The introduction should be expanded to include a more comprehensive discussion of stability analysis in wheat breeding.
Key studies related to genotype-environment interaction and stability indices should be reviewed to provide better context for the study.

Experimental design

Absence of a Check Variety:
The manuscript does not include a check variety for comparison, which is a critical component of genotype evaluation. The authors need to justify this omission or, if possible, include a check variety in the study.
Source of Genotypes:
The manuscript does not mention the source of the genotypes used. It is essential to specify where the genotypes were obtained from—whether from a breeding program, national gene bank, or other sources.
Methodological Clarity and Justification:
The methodology section lacks sufficient detail on stability indices and why specific indices were chosen.
The statistical methods, including the stability parameters used, need more justification. The authors should explain why these particular methods were applied over others commonly used in genotype evaluation.

Validity of the findings

Biplot Analysis and AMMI Model:
The manuscript discusses biplot analysis but does not provide enough explanation on the AMMI model. The authors should elaborate on how AMMI contributes to stability analysis and why it is advantageous in assessing genotype performance.
Over-Reliance on a Single Author in References:
The reference list contains excessive citations of Al-Ashkar I (eight times). A broader literature review incorporating studies from various researchers is necessary to strengthen the scientific foundation of the study.
Additionally, references are not formatted correctly. The authors should ensure adherence to the journal’s reference formatting guidelines.
Results and Discussion Need More Depth:
The discussion should better relate findings to previous studies and theoretical frameworks. How do these results compare to other stability studies in wheat?
The practical implications of the findings should be elaborated upon, particularly in relation to wheat breeding programs and climate resilience.

Additional comments

The manuscript cannot be accepted in its present form and requires major revision. The authors need to substantially improve the introduction, justify their methodology, clarify missing details (such as the source of genotypes and the absence of a check variety), strengthen the discussion on stability analysis, and improve the overall structure and formatting of references. Once these major issues are addressed, the manuscript may be reconsidered for further review.

Reviewer 2 ·

Basic reporting

This study aimed to identify the optimal genotypes that combine stability and high productivity to confront thermal stress ii) validate the proficiency of 18 selection indices used in screening tolerant genotypes via a variety of statistical approaches iii) assess the associations among the different indices. The language of this study is not understandable, and grammatically quite good. Therefore, the manuscript does need to be edited in terms of language. The study needs minor grammatical corrections. Only long sentences shoul be shortened. The abstract, introduction, material methods, results and discussions, and conclusion parts of the manuscript are written very well. Also, the discussion section should be enriched with new literature. So, in the study, there are not important points that are overlooked. Other and specific comments are given in the text. Also, the references of the study were not checked again and the suitability of the journal format was left to the authors.

Experimental design

It is used when there is a factor that disrupts the homogeneity of the block material. Which factor did you take as a block in the statistical model? It would be useful if you could explain.

The statistical model should be checked. It is useful to explain the logic of using the nested structure in the second source of variation.

If the study was conducted with 3 replications, how was a multivariate analysis performed? In the literature, 10 times the number of replications is required to perform a multivariate analysis. Explain how you performed this analysis.

Validity of the findings

The most beautifully expressed section of the study is the results section, which is written quite impressively.

Annotated reviews are not available for download in order to protect the identity of reviewers who chose to remain anonymous.

Reviewer 3 ·

Basic reporting

The manuscript is written nicely, but at some places, minor revisions are required

Experimental design

Statistical analysis complements the research findings

Validity of the findings

Finding of result reflects valid and authentic

Additional comments

Some minor revisions are needed as follows:
Kindly check the uniformity of citations/references
Include the GPS data for locations
Recast the sentences already highlighted in the reviewed manuscript

Annotated reviews are not available for download in order to protect the identity of reviewers who chose to remain anonymous.

---

## Round 0.2 · Minor Revisions

Dear Authors

The manuscript still needs a minor revision before publication. The authors are invited to revise the paper, taking into account all the suggestions made by the reviewers. Please note that the requested changes are required for publication.

With Thanks

Reviewer 2 ·

Basic reporting

The necessary revisions have been made carefully in the study. It is believed that the final version of the manuscript will contribute to the literature. Therefore, I wholeheartedly congratulate the authors.

Experimental design

The necessary revisions have been made carefully in the study.

Validity of the findings

The necessary revisions have been made carefully in the study.

Additional comments

The study can be published in its final form.

Reviewer 3 ·

Basic reporting

Language is clear and self explanatory
Some typographic errors are there.
References are sufficient to justify manuscript

Experimental design

Experimental design is up to the mark

Validity of the findings

ok

Annotated reviews are not available for download in order to protect the identity of reviewers who chose to remain anonymous.

Reviewer 4 ·

Basic reporting

no comment

Experimental design

no comment

Validity of the findings

no comment

Additional comments

This study effectively addresses the critical issue of thermal stress on wheat, a significant threat to food security exacerbated by climate change. It highlights the complexities of genotype-environment interaction (GEN:ENV) and successfully identifies stable, high-yielding wheat genotypes with improved heat tolerance. The research employs robust methodologies, including stability analyses and selection indices, which offer valuable tools for breeders. Ultimately, the findings provide crucial insights for developing heat-resilient wheat cultivars, directly supporting efforts to ensure future food supplies.
-Comments and Suggestions for Authors
The manuscript is good, but some English phrasing hinders comprehension. I worked to improve the scientific writing throughout the manuscript.
1- Line 45: "due to a miscalculation in genetics" Rephrase for clarity; perhaps "due to the complex genetic basis of yield under varying conditions" or "due to the intricate genetic architecture involved."
2- Line 54: "three (G05, G09, and G17) of them showed more stable." Should be "three (G05, G09, and G17) were more stable."
3- Line 56: "and selected too as the best genotypes group by WAASB-GY, except G18." Rephrase for conciseness and clarity, e.g., "and were also identified as the best genotype group by WAASB-GY, with the exception of G18."
4- Line 59-60: "which showed that the genotypic effect plays a major role in their inheritance, except for the SNPI index." Rephrase for better flow, e.g., "indicating a major role of the genotypic effect in their inheritance, with the exception of the SNPI index."
5- Line 75: "which negatively affects the grain size and quality" Consider rephrasing as "negatively affecting grain size and quality."
6- Line 88-89: "This is something undesirable at the time due to winter crops being adversely affected by warmer which determines many yield-contributing traits..." Rephrase to "This is undesirable as warmer conditions adversely affect winter crops, impacting many yield-contributing traits..."
7- Line 97-98: "A plant's ability to transcend thermal stress is appropriate circumstances, agronomy, and genetic factors that improve evaporative cooling potential" Rephrase to "A plant's ability to overcome thermal stress depends on appropriate environmental conditions, agronomic practices, and genetic factors that enhance evaporative cooling potential."
8- Line 100-101: "to obtain high-yielding model varieties, combining productive and thermal tolerant" Should be "combining productivity and thermal tolerance."
9- Line 103-104: "the genotype performance is different from strength to weakness or conversely in various seasons" Rephrase to "genotype performance varies from superior to inferior or vice-versa across different seasons."
10- Line 112: "must be able to genotypes distinguish" Should be "must be able to distinguish genotypes."
11- Line 114-115: "depends on the intensity of environmental stress varies throughout years and regions, affecting how well selection indices identify tolerant genotypes" Rephrase to "depends on the intensity of environmental stress, which varies across years and regions, thereby affecting the efficacy of selection indices in identifying tolerant genotypes."
12- Line 118-119: "indicated that multiple studies have highlighted the efficiency in selecting the indices tolerance, but did not fully address it because of their dependence on simple statistics." Rephrase to: "indicated that while multiple studies have highlighted the efficiency of selection indices for tolerance, these studies did not fully address it due to the indices' dependence on simple statistics."
13- Line 122-123: "which may assist show favorite genotypes” Should be "which may help identify favorable genotypes."
14- Line 132: "recognize the genotypes high-yielding and stable" Should be "recognize the high-yielding and stable genotypes."
15- Line 136: "the main limitation was noted when analyzing the structure of the linear mixed-effect model (LMM), so, novel model" Should be "a main limitation was noted when analyzing the structure of the linear mixed-effect model (LMM); therefore, a novel model..."
16- Line 143-144: "achieving the same goal to discriminate the GEN: ENV pattern from the random error, but they are statistically different" Rephrase to: "achieving the same goal of discriminating the GEN:ENV pattern from random error, despite being statistically different."
17- Line 162: "seedling rate, fertilizing rates, and at the time of add, and meteorological conditions" "at the time of add" is unclear. Clarify what was added.
18- Line 202-203: "estimating relationships between the various indices such as (genetic (rg) and phenotypic (rp) correlations, genetic parameters, and MGIDI index)." Rephrase to "estimating relationships between the various indices, including genetic (rg) and phenotypic (rp) correlations, genetic parameters, and the MGIDI index."
19- Line 218: "grossed the most" – While understandable, consider more formal scientific language like "achieved the highest yield" or "performed best."
20- Line 222: "Joint ANOVA and AMMI model analyses to grain yield" – "to grain yield" should be "for grain yield."
21- Line 418-419: "the adaption map showed that G05, G09, and G17 were better suited and exhibited similar performance in the ENVs" "adaption map" should be "adaptation map."
22- Line 427: "reduces redundancy accounts" "reduces redundancy" is sufficient.
23- Line 431-432: "where higher ranks are represented by darker hues and lower rankings by lighter hues" – "rankings" should be "ranks."
24- Line 432-433: "G01, G02, G03, G04, G05, G06, and G09 showed the lowest WAASB values (so they were more stable and performed)" "performed" is incomplete. Should be "performed well" or "showed higher performance."
25- Line 458-459: "so it will be easy to familiarize high-performance genotypes from GYoc and GYtsc (the outcome of GYoc to acquire the most effective selection GYtsc indirectly)." This sentence is very difficult to understand. It needs significant rephrasing for clarity.
26- Line 460: "RDC had a negative and positive correlation with GYtsc and GYoc, respectively;" The "negative and positive" needs clarification as to which index correlates negatively with which GY and positively with which GY.
27- Line 467-468: "using cutting-edge statistical techniques to better understand genetic factors and identify indices that are least impacted by the environment." "impacted" could be "influenced" or "affected."
28- Line 474-475: "the interaction was simple, therefore, undesirable low-value for choosing genotypes" – Rephrase for clarity.
29- Line 479: "CVs (g/r) ratio was greater than 1, showing little variation" This statement seems contradictory based on typical CV interpretation (higher CV = more variation). Clarify if this is specific to their context or if "little variation" refers to something else.
30- Line 493: "A distinct-to-understand selection process unique" Rephrase to "A distinct and easy-to-understand selection process..."

---

## Round 0.3 · accepted · Accept

Dear Authors,

I am pleased to inform you that the manuscript has been improved following the last revision and can now be accepted for publication.

Congratulations on accepting your manuscript. Thank you for your interest in submitting your work to PeerJ.

With Thanks

Reviewer 4 ·

Basic reporting

no comment

Experimental design

no comment

Validity of the findings

no comment

Additional comments

The authors have made the changes I suggested in the last review. I recommend its publication in this journal.